# Escaping the big data paradigm in self-supervised representation learning

## Abstract

The reliance on large-scale datasets and extensive computational resources has become a significant barrier to advancing representation learning from images, particularly in domains where data is scarce or expensive to obtain. In this paper, we address the critical question: *Can we escape the big data paradigm in self-supervised representation learning from images?* We introduce **SCOTT** (**S**parse **Co**nvolutional **T**okenizer for **T**ransformers), a simple tokenization architecture that injects convolutional inductive biases into Vision Transformers (ViTs), enhancing their efficacy in small-scale data regimens while remaining compatible with Masked Image Modeling (MIM) tasks. Alongside, we propose **MIM-JEPA**, a Joint-Embedding Predictive Architecture within a MIM framework, operating in latent representation space to capture more semantic features. Our approach enables ViTs to be trained from scratch on datasets orders of magnitude smaller than traditionally required –without relying on massive external datasets for pretraining. We validate our method on three small-size, high-resoultion, fine-grained datasets: Oxford Flowers-102, Oxford IIIT Pets-37, and ImageNet-100. Despite the challenges of limited data and high intra-class similarity, our frozen SCOTT models pretrained with MIM-JEPA significantly outperform fully supervised methods and achieve competitive results with state-of-the-art approaches that rely on large-scale pretraining, complex image augmentations and bigger model sizes. By demonstrating that robust off-the-shelf representations can be learned with limited data, compute, and model sizes, our work paves the way for computer applications in resource constrained environments such as medical imaging or robotics. Our findings challenge the prevailing notion that vast amounts of data are indispensable for effective representation learning, offering a new pathway toward more accessible and inclusive advancements in the field.

## 1 Introduction

Escaping the big data paradigm in self-supervised learning from images is crucial for the future of computer vision (CV). Representation learning, described in (Bengio et al., 2013) as "learning representations of the data that make it easier to extract useful information when building classifiers or other predictors", becomes particularly relevant when training data is scarce as it would enable efficient learning for downstream tasks. Traditionally, transfer learning has been the dominant approach, where convolutional neural networks (CNNs) (LeCun et al., 1989) are pretrained on large-scale labeled datasets like ImageNet (Deng et al., 2009) and then fine-tuned on specific tasks. However, this approach has two major constraints: the reliance on vast labeled datasets for pretraining and the domain-specific brittleness of the learned features (Jain et al., 2023). These limitations make it impractical in fields like medical imaging or industrial applications, where data collection requires domain-expertise and is both time-consuming and expensive (Huang et al., 2023).

In recent years, self-supervised learning (SSL) has emerged as a promising alternative, motivated by the success of methods such as BERT (Devlin et al., 2019) and GPT (Radford et al., 2019) in natural language processing (NLP). The core idea behind SSL is to devise a task that provides a supervisory signal from the data itself without explicit human annotation, allowing models to learn meaningful representations in a label-free environment (Caron et al., 2021). However, self-supervised learning success in both NLP and CV must largely be attributed to the advent of the Transformer architecture (Vaswani, 2017), which leverages self-attention mechanisms to capture long-range dependencies in

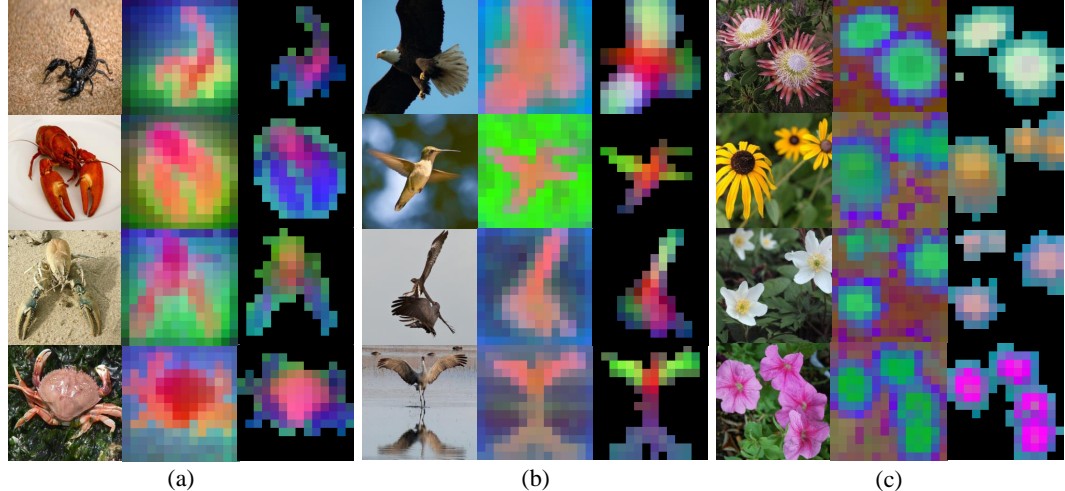

(a)                (b)                (c)

Figure 1: **Matching different semantic parts across categories and poses**. We show the first 3 components of a PCA computed among the token embeddings of images from the same column (a, b, and c). The background is removed by thresholding the first component. Notably, semantically similar parts are matched by color despite belonging to different object classes and poses. For instance: in (a) animal claws are purple and torso pink, in (b) wings are green and torso red. Interestingly, once background is removed in (c), different flower disks are matched to different colors.

data in a highly parallel and scalable manner. The Vision Transformer (ViT) (Dosovitskiy, 2020) marked the first significant attempt to apply a purely transformer architecture to visual tasks, but its success hinges on access to extremely large datasets (14M-300M images) (Deng et al., 2009; Sun et al., 2017; Asano et al., 2021). As ViT authors noted, Transformers lack certain inductive biases inherent to CNNs -such as translation equivariance and locality- which makes them less effective when trained on limited data (Dosovitskiy, 2020).

Over the past few years, this combination of label-free training methods with ViT has led to a "resource-hungry" training paradigm, with most research efforts pushing the state of the art in the natural image domain through scaling to even larger models and dataset sizes. Unfortunately, this trend limits major contributions from researchers with limited compute and data budgets and poses significant challenges in specialized fields where domain-specific data is difficult to acquire. Therefore, escaping the big data paradigm is crucial for advancing computer vision applications in fields beyond natural images. By reducing the dependency on large datasets, we could make advancements in this field more accessible and impactful across a wider range of applications (Huang et al., 2023).

Thus, a pressing question arises: **Can we escape the big data paradigm in self-supervised representation learning from images?**

In this work, we take a step towards addressing this challenge by introducing two key contributions: the **S**parse **Co**nvolutional **T**okenizer for **T**ransformers **(SCOTT)** and a Joint-Embedding Predictive Architecture (JEPA) (LeCun, 2022) for vision instantiated in a Masked Image Modeling (MIM) framework (Bao et al., 2021), which we refer to as **(MIM-JEPA)**. SCOTT is a tokenization architecture that replaces the original patch-based embedding of ViTs, and not only incorporates the inductive biases of CNNs to allow ViT to operate effectively in small-scale data regimes, but also its sparsity mitigates issues like information leakage and mask vanishing, which have previously hindered the application of MIM strategies in CNN-based tokenizers for transformers. Moreover, in contrast to generative MIM methods that predict missing information in pixel/token space, the JEPA objective is in abstract representation space where unnecessary pixel-level details are potentially eliminated, leading the model to produce more semantic features. This capability is demonstrated in Figures 1 and 3, where we apply a principal component analysis (PCA) on the patch features produced by our method, revealing meaningful semantic structures.

To prove our method's potential to unlock deep learning for the long tail of vision tasks without expensive labelled datasets, we constrain our research to three small-size, high resolution, fine-grained

datasets. Specifically, we focus on two popular computer vision datasets from the VTAB benchmark (Zhai et al., 2019): Oxford Flowers-102 (Nilsback & Zisserman, 2008) and Oxford IIIT Pets-37 (Parkhi et al., 2012); the third one is ImageNet-100 (Deng et al., 2009), a subset of the well-studied ImageNet with 100 different classes of animals. Apart from the small data available for training, with roughly 20 samples per class in Flowers-102, these datasets present a significant challenge due to their high intra-class similarity, making them ideal for testing the limits of self-supervised learning without large datasets. It is worth noting that unlike previous works (Dosovitskiy, 2020; Bao et al., 2021; Assran et al., 2023; Oquab et al., 2024; Zhou et al., 2021), (Steiner et al., 2021), that rely on pretraining on massive external datasets for learning to see (Steiner et al., 2021), our method is trained entirely from scratch using only the images, without labels, of the target dataset.

In summary, our contributions are as follows:

- We propose SCOTT, a **S**parse **Co**nvolutional **T**okenizer for **T**ransformers that incorporates CNN-like inductive biases within ViTs and is compatible with MIM training due to its sparse architecture.
- We introduce a self-supervised learning framework based on a Joint-Embedding Predictive Architecture (JEPA) instantiated in a MIM task, referred to as MIM-JEPA, which enhances performance on fine-grained visual tasks.
- We demonstrate that combining SCOTT and a JEPA enable Vision Transformers to perform effectively in small-scale environments, significantly outperforming previous methods and drastically reducing reliance on large datasets.
- Our method is accessible to researchers with limited computational resources, thereby making state-of-the-art self-supervised learning more inclusive and adaptable across fields.

Through this work, we aim to advance self-supervised learning in computer vision by making it more accessible and practical for a broader spectrum of applications, particularly in domains where large-scale datasets are not feasible. We present an efficient model with few parameters, that can be quickly and effectively trained on smaller platforms while still maintaining state-of-the-art results.

## 2 RELATED WORKS

### 2.1 INJECTING ViT WITH CONVOLUTIONAL PRIORS

Vision Transformer (ViT) reliance on large datasets stems from the lack of inductive biases inherent to convolutional neural networks (CNNs) (Dosovitskiy, 2020). CNNs, inspired by the hierarchical processing of the mammalian visual cortex (Hubel & Wiesel, 1959; Fukushima, 1988), provide important priors for learning spatial relationships in visual data. Recognizing this limitation, numerous studies have previously explored incorporating convolutional priors into ViT architectures (Wu et al., 2021; Chen et al., 2021; Yuan et al., 2021; Graham et al., 2021). Early attempts, such as the "hybrid ViT" (Dosovitskiy, 2020), fed a ResNet (He et al., 2016) feature map into a transformer encoder, showing a slight performance advantage over ViT at smaller computational budgets. However, later studies (Xiao et al., 2021) revealed that excessive convolutional layers could diminish the generalization power of ViTs, suggesting that a shallow convolutional stem might strike the right balance between CNN-like inductive biases and the representational power of transformers.

The Compact Convolutional Transformer (CCT) (Hassani et al., 2021) follows this principle, introducing a convolutional tokenizer for supervised learning on datasets significantly smaller than ImageNet. While CCT focuses on supervised training, our work pushes the idea further by leveraging sparse convolutions (Liu et al., 2015) –following pioneering work of (Tian et al., 2023) to enable BERT pre-training on CNN architectures– to enhance tokenization specifically for self-supervised learning. This sparse convolutional architecture overcomes critical limitations such as information leakage and mask vanishing that hampered the application of traditional convolution-based tokenizers for ViTs in MIM tasks until now. We refer the reader to (Tian et al., 2023) for further analysis.

### 2.2 MASKED PREDICTIVE REPRESENTATION LEARNING

Masked Image Modeling (MIM), first introduced in BEiT (Bao et al., 2021), draws inspiration from the success of BERT in NLP (Devlin et al., 2019). In this approach, an image is divided in non-

overlapping patches and a subset of these patches is masked out. The model is tasked with reconstructing the masked regions, which encourages learning meaningful representations of visual features, akin to how BERT learns semantic dependencies in text. Since the introduction of MIM, various methods have explored different reconstruction targets, such as raw pixels (He et al., 2022; Xie et al., 2020; 2022), or patch-level tokens via a learned tokenizer (Bao et al., 2021; Peng et al., 2022). While these approaches have been effective in scaling self-supervised learning to larger datasets, they often lead to feature representations at a low-level of semantic abstraction. This is particularly problematic in fine-grained tasks, which require deeper more abstract representations for distinguishing between visually similar classes. Invariance-based pretraining methods that enforce similar embeddings for two or more views of the same image (Zhou et al., 2021; Oquab et al., 2024) have been combined with MIM objectives, to produce representations of a high semantic level. However, image views are typically constructed using a set of complex hand-crafted data augmentations that introduce strong biases that may be detrimental to certain downstream tasks (Assran et al., 2023) and also may not generalize to other scientific domains (Huang et al., 2023).

Our work is directly inspired by I-JEPA (Assran et al., 2023), which takes this concept further by predicting masked abstract targets in the representation space produced by a momentum-based target-encoder ViT network. Building on these ideas, we integrate our Sparse Convolutional Tokenizer for Transformers (SCOTT) within the ViT architecture of a JEPA framework based on MIM and dubbed MIM-JEPA. This combination enables effective self-supervised learning on small-scale datasets, where traditional ViT approaches typically struggle.

Our work differs from the aforementioned works in several ways, in that it focuses on proposing a learning framework and a model that can be trained from scratch on small datasets that are orders of magnitude smaller than ImageNet. Thus, offering a solution to efficiently train models, with fewer parameters, on small datasets and smaller platforms while still maintaining state-of-the-art results.

## 3 METHOD

To provide empirical evidence that ViTs can be effectively trained from scratch on small datasets, we propose to harness the full power of self-supervised learning for learning representations. To this end, we design a Joint-Embedding Predictive Architecture instantiated through a MIM task, referred to as MIM-JEPA, and illustrated in Figure 2.

The overall training objective is as follows: given a masked image as input to a context-encoder, a predictor is tasked with learning the latent representations of the masked blocks produced by a target-encoder that processes the full image. Furthermore, to address the suboptimal optimizability of ViTs caused primarily by the *patchify* stem (i.e., tokenizer), we propose to replace it by a Sparse Convolutional Tokenizer for Transformers (SCOTT). This tokenizer is compatible with MIM objectives and introduces convolutional priors into ViTs, offering superior data efficiency and performance.

### 3.1 SPARSE CONVOLUTIONAL TOKENIZER FOR TRANSFORMERS (SCOTT) ARCHITECTURE

A standard transformer (Vaswani, 2017) takes as input a sequence of vectors, called tokens. However, there is a fundamental difference between the signal space of NLP and the signal space of computer vision, given that language data is discrete and structured (i.e., words), whereas image content is high dimensional, continuous, and unstructured (i.e., pixel values) (Ozbulak et al., 2023).

Image tokenization in standard ViTs is performed by a patch and embed layer which subdivides an image into non-overlapping square patches so that a transformer can accept visual data. Formally, the image $x \in R^{H \times W \times C}$ is reshaped into $N = HW/P^2$ patches $x^p \in R^{H \times (P^2 C)}$, where $C$ is the number of channels, $H, W$ is the input image resolution, and $(P, P)$ is the resolution of each patch. The image patches $\{x_i^p\}_{i=1}^N$ are then linearly projected into patch embeddings $\{e_i^p\}_{i=1}^N$ each with dimension $d$. This is equivalent to a convolutional layer with $d$ filters, and $P \times P$ stride and kernel size. Among other limitations, this simple patch and embedding method eliminates the boundary-level information present in different patches.

Specifically in our experiments, we split each $224 \times 224$ image into a $14 \times 14$ grid of patch embeddings, where each embedding corresponds to a $16 \times 16$ image patch.

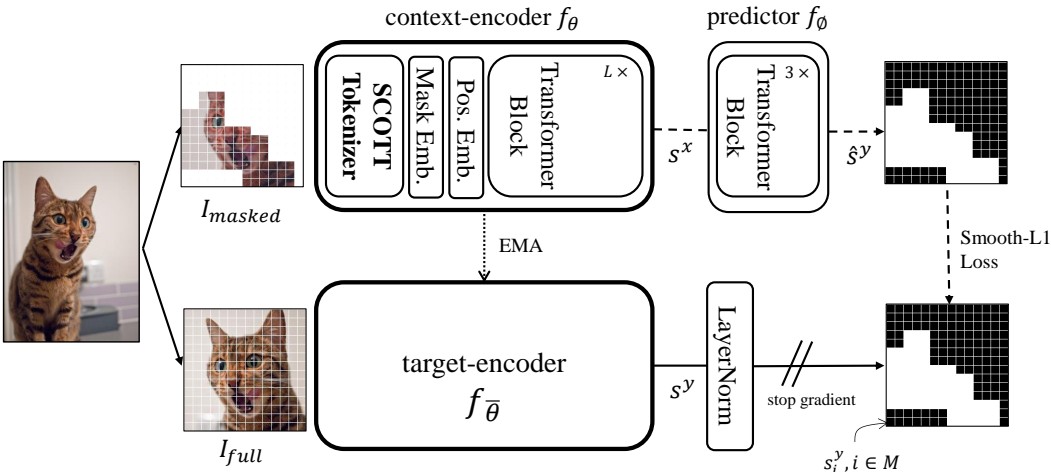

Figure 2: **MIM-JEPA.** An image $I_{full}$ is processed by the target-encoder $f_{\bar{\theta}}$ to produce a latent patch-level representation $s^y$, whose masked patches $M$ are used as targets; The context image $I_{masked}$, generated from the complement of $M$, is input to the context-encoder $f_{\theta}$ to produce $s^x$. The predictor $f_{\phi}$ is fed with $s^x$ to predict the missing content $\hat{s}^y$. The Smooth-L1 loss is computed only on the (black) masked patches in latent space to update the context-encoder and predictor weights (dashed line), while the target encoder's weights are updated via an exponential moving average (EMA) of the context-encoder (dotted line).

In order to inject some inductive biases into the transformer architecture, we propose to replace the patch and embedding in ViT by a shallow convolutional stem. This stem follows conventional design, which consists of 2 consecutive blocks of: convolution, ReLU activation, and a max blur pool layer (Zhang, 2019) (see Appendix A.1 for details). The output of the convolutional stem proposed produces $\{e_i^p\}_{i=1}^N$, a $14 \times 14$ feature map each with dimension $d$ matching the number of inputs to the transformer created by the standard patch and embedding method.

However, introducing a CNN tokenizer conflicts with the patch-wise masking strategy because one cannot eliminate pixel information from masked patches -to avoid trivial solutions- as ViTs do by removing or replacing them with a mask token. Setting masked patches to zero in CNNs has drawbacks: (i) it disturbs the pixel value distribution; (ii) masked patterns vanish after several convolutional layers; (iii) computations on masked regions are unnecessary. To overcome this, inspired by SparK (Tian et al., 2023), we gather masked patches into a sparse image and employ sparse layers that compute only when the kernel center covers a non-empty element (see "submanifold sparse convolution" in (Graham & Van der Maaten, 2017)). Since dense images are special cases of sparse images without holes, sparse layers naturally reduce to standard ones when masking isn't applied.

**SCOTT enabled Vision Transformer.** Following ViT (Dosovitskiy, 2020), our backbone network is a standard Transformer (Vaswani, 2017) to ensure a fair comparison between our results and previous works in terms of network architecture. Specifically, our ViT can be decomposed in parts: SCOTT for image tokenization, fixed sinusoidal Positional Embedding, a Mask Token, and $L$ consecutive Transformer Encoder blocks. Since our method learns representations without labels, we do not use a class token nor a classification head present in the standard ViT. The features used in downstream tasks are the model's frozen output. We use similar notation in ViT for SCOTT enabled variants: for instance, SCOTT-7/16 is a vision transformer that has a SCOTT tokenizer with a patch size of 16 and 7 transformer encoder blocks.

### 3.2 LEARNING IMAGE REPRESENTATIONS (MIM-JEPA)

We first formally describe Masked Image Modeling (MIM) which lays the foundation for then proposing the MIM-JEPA learning framework, which allows to instantiate a Joint-Embedding Predictive Architecture in the context of images using masking.

**Masked Image Modeling:** an input image is first tokenized into patch embeddings $\{e_i^p\}_{i=1}^N$, as explained in Section 3.1. Following that, a portion of the patch embeddings is selected to be masked. Denoting the masked position set as $M$, a shared learnable embedding $e_M$ replaces the original patch embeddings $e_i^p$ when $i \in M$, producing the masked sequence:

$$e_i^M = \delta(i \in M) \odot e_M + (1 - \delta(i \in M)) \odot e_i^p \tag{1}$$

where $\delta(\cdot)$ is the indicator function. Subsequently, the positional embedding is added and then fed the sequence into the $L$ transformer encoder blocks. After that, the output vectors $s = \{s_i\}_{i=1}^N$ are regarded as the encoded semantic representations of the input image patches. Thus, $s_i$ is the representation associated with the $i^{th}$ patch.

**Learning Image Representations in a Joint-Embedding Predictive Architecture through Masked Image Modeling (MIM-JEPA).** JEPAs are conceptually close to Generative Architectures, however, the loss function is applied in embedding space, not input space. The overall training objective is as follows: given a masked image as context to a context-encoder, task a predictor to learn the latent representations of the masked patches of the image produced by a target-encoder that is fed with the full image. We use a SCOTT enabled ViT, introduced in Section 3.1, for the context-encoder $f_\theta$ and target-encoder $f_{\bar\theta}$, the predictor $f_\phi$ is a shallow standard transformer (Vaswani, 2017) that takes as input the context-encoder outputs. Following we describe how we produce each of the MIM-JEPA components: masking, targets, context, prediction and loss, given an input image.

**Masking.** In order to generate the masks for our MIM objective, we follow previous work to use a Blockwise masking strategy (Bao et al., 2021). Specifically, given an input image, we iteratively sample possibly overlapping blocks with random aspect ratio until enough patches are masked in $M$. In our experiments, $0.6N$, where $N$ is the total number of patches and $0.6$ the masking ratio. This masking strategy produces masked context-images that are informative and target-patches that are relatively semantic. See the masked image, $I_{masked}$, in Figure 2.

**Targets.** In the MIM-JEPA framework, the targets correspond to the latent representations of image blocks $s^y = \{s_i^y\}_{i=1}^N$ produced by the target-encoder $f_{\bar\theta}$ fed with the full input image, $I_{full}$. Thus, once $s^y$ is available, the target blocks are obtained by masking $s^y$ instead of the input image.

**Context.** Similarly, the masked input image $I_{masked}$, i.e., the image with patch-size holes, see Figure 2, is fed into the context-encoder network $f_\theta$ to produce the corresponding patch-level representation $s^x = \{s_i^x\}_{i=1}^N$.

**Prediction.** Since the goal behind JEPAs is to predict the representations in an embedding space, we feed the context patch-level representations $s^x$ to the predictor $f_\phi$ which outputs the corresponding patch-level predictions $\hat{s}^y = \{\hat{s}_i^y\}_{i=1}^N$.

**Loss.** The loss $L$ is simply the Smooth-L1 loss over the predictions $\hat{s}^y$ and the $N$ layer normalized (Lei Ba et al., 2016) features $s^y$ produced by the target-encoder $f_{\bar\theta}$. Importantly, the loss is only applied to the masked patches to encourage the model to learn patch-level representations that are predictive of each other; predicting non-masked patches is trivial.

The full training objective can be unified as:

$$MIM = L(f_\phi(f_\theta(I_{masked})), N(f_{\bar\theta}(I_{full}))) \tag{2}$$

The parameters of the context-encoder, $\theta$, and the predictor, $\phi$, are jointly learned via gradient-based optimization, while the target-encoder's parameters, $\bar\theta$, are updated via an exponential moving average (EMA) of the context-encoder parameters. Using an EMA target-encoder, an asymmetric architecture between the $x$- and $y$- encoding paths, and the layer normalization over target features $s^y$ has proven to avoid representation collapse and help training in previous works (Assran et al., 2023; Grill et al., 2020; Geiping et al., 2023), the same holds true for MIM-JEPA.

**Image augmentations.** Drawing inspiration from view-invariant SSL methods, we try to induce a shape-bias –a property of human perception (Naseer et al., 2021)– by randomly applying a set of simple image transformations: color jitter, grayscale, and gaussian blur, to a given input image to produce two views with slightly different color properties while preserving spatial content.

## 4 EXPERIMENTS

### 4.1 DATASETS

Recall that our objective is to develop a method capable of efficiently training from scratch on small-sized, high-resolution, fine-grained datasets, while still maintaining state-of-the-art results. In that sense, we focus on 3 datasets, 2 popular computer vision datasets from the VTAB benchmark (Zhai et al., 2019): Oxford Flowers-102 (Nilsback & Zisserman, 2008) and Oxford IIIT Pets-37 (Parkhi et al., 2012), and the ImageNet-100 (Deng et al., 2009). We selected these datasets for several reasons: (i) they are all considered small-sized datasets in literature with a huge gap in top-1 accuracy between from scratch training and large-scale pretrained models (Steiner et al., 2021), (ii) they are all high-resolution, i.e., $224^2$ images. (iii) Flowers-102 and Pets-37 present a significant challenge due to their high intra-class similarity. (iv) ImageNet-100 is a subset of ImageNet which contains 100 different classes of animals. (v) The Magnitude of the image-per-class ratio for supervised training increases across the datasets, where $ratio = \frac{I_{train}}{N_{classes}}$. Further details in Appendix B.

### 4.2 SELF-SUPERVISED PRETRAINING (SCOTT + MIM-JEPA)

In contrast to SL, which requires labeled datasets, our MIM-JEPA pretraining strategy enables models to harness the full power of unsupervised learning paradigms by learning representations directly from the data itself, without labels. Leveraging this property, as more data yields more generic features (see Table 11), we use the full unlabeled target dataset during MIM-JEPA pretraining.

**Optimization**. All models are trained at $224 \times 224$ input resolution. We use AdamW (Loshchilov, 2017) to jointly optimize the context-encoder and predictor with a batch size of 128, fitting in a single NVIDIA RTX 3090 GPU. For the learning rate, we follow a explore-exploit schedule (Iyer et al., 2023) with a linear warmup to its peak value of $5e - 4$, a flat *explore* phase for $0.72$ of the remaining epochs and a final *exploit* phase with a cosine decay schedule. Weight decay is linearly increased from $0.04$ to $0.4$. For the target-encoder, the EMA parameter starts at $0.996$ and is linearly increased to $1$ during training. All hyperparameters are summarized in Appendix D.

### 4.3 DOWNSTREAM TASK: FROZEN IMAGE CLASSIFICATION

To demonstrate that our method learns highly semantic representations during MIM-JEPA pretraining, we present results on transferring the learned frozen features to image classification tasks. We focus on classification because many industrial and medical applications rely on classification (e.g., disease or defect detection); thus, our research may be well-suited for them. (Huang et al., 2023)

**Evaluation**. After self-supervised pretraining (MIM-JEPA) on the unlabeled target dataset for 300 epochs following Section 4.2, the model weights are frozen, and a simple, lightweight classifier is trained on top for 100 epochs using only the training split in a supervised manner. The images are resized to $256^2$ pixels from which a $224^2$ center crop is extracted. For all datasets we report Top-1 and Top-5 classification accuracy as our main metrics. Consistent with previous work (Bardes et al., 2024), we find that attentive-probing achieves better results, although linear-probing is still feasible.

As shown in Table 1, our MIM-JEPA self-supervised pretraining drastically improves performance across all tested datasets and architectures compared to models trained from scratch using only the target dataset and fully supervised learning. For example, on the Pets-37 dataset, a ViT-12/16 trained from scratch achieves a Top-1 accuracy of 48.3%, whereas an attentive probe on top of a frozen SCOTT-12/16 model pretrained with MIM-JEPA attains a Top-1 accuracy of 90.7%, representing a significant increase of 42.4 percentage points. Additionally, SCOTT-enabled ViTs outperform the standard ViT architecture. Notably, the performance achieved by frozen SCOTT models pretrained with MIM-JEPA is on par with ViT models pretrained on large-scale datasets and fine-tuned on the target dataset. For instance, on the Flowers-102 dataset, our frozen SCOTT-7/16* model with 14 million (M) parameters achieves a higher Top-1 accuracy (96.9%) than a ViT-12/16 (95.7%) with 22 M parameters pretrained on ImageNet-1k (1.3 M images), despite our SCOTT model being pretrained using only 8,189 unlabeled images.

Furthermore, we asses the performance of SCOTT models with MIM-JEPA pretraining against state-of-the-art self-supervised transformer methods, such as DINOv2 (Oquab et al., 2024) and I-JEPA

Table 1: Comparison of our method in Top-1 and Top-5 accuracies (%) to different methods across different datasets. Notably, SCOTT models pretrained using MIM-JEPA achieve competitive performance with state-of-the-art models, despite being pretrained exclusively on the unlabeled target dataset—which is orders of magnitude smaller and less heterogeneous. SCOTT models marked with an asterisk (*) were pretrained for longer (1200 epochs instead of 300).

| Model | | Pretraining strategy | | | Downstream SL | |
|---|---|---|---|---|---|---|
| Name | #Params | Method | Dataset | #Samples | Top-1 | Top-5 |
| **Oxford Flowers-102** | | | | | | |
| ViT-12/16 | 22 M | - | - | - | 71.1 | 87.5 |
| SCOTT-7/16 | 14 M | - | - | - | 79.1 | 92.2 |
| SCOTT-12/16 | 22 M | - | - | - | 79.1 | 91.9 |
| Fine-tuned ViTs from supervised pretraining (SL) | | | | | | |
| ViT-12/16 | 22 M | SL | ImageNet-1k | 1.3 M | 95.7 | - |
| ViT-12/16 | 22 M | SL | ImageNet-21K | 14.2 M | 99.6 | - |
| Self-supervised learning pretrained ViTs | | | | | | |
| ViT-12/14 + reg | 22 M | DinoV2 | LVD-142M | 142.0 M | 99.6 | 99.9 |
| ViT-32/14 | 630 M | I-JEPA | ImageNet-1k | 1.3 M | 93.7 | 98.5 |
| MIM-JEPA pretrained SCOTT enabled ViTs (ours) | | | | | | |
| SCOTT-7/16 | 14 M | MIM-JEPA | Flowers-102 | 8189 | 95.7 | 99.0 |
| SCOTT-7/16* | 14 M | MIM-JEPA | Flowers-102 | 8189 | 96.9 | 99.3 |
| SCOTT-12/16 | 22 M | MIM-JEPA | Flowers-102 | 8189 | 97.1 | 99.1 |
| SCOTT-12/16* | 22 M | MIM-JEPA | Flowers-102 | 8189 | 97.7 | 99.2 |
| **Oxford IIIT Pets-37** | | | | | | |
| ViT-12/16 | 22 M | - | - | - | 48.3 | 78.5 |
| SCOTT-7/16 | 14 M | - | - | - | 67.3 | 89.3 |
| SCOTT-12/16 | 22 M | - | - | - | 67.5 | 90.2 |
| Fine-tuned ViTs from supervised pretraining (SL) | | | | | | |
| ViT-12/16 | 22 M | SL | ImageNet-1k | 1.3 M | 93.8 | - |
| ViT-12/16 | 22 M | SL | ImageNet-21K | 14.2 M | 93.2 | - |
| Self-supervised learning pretrained ViTs | | | | | | |
| ViT-12/14 + reg | 22 M | DinoV2 | LVD-142M | 142.0 M | 94.8 | 99.9 |
| ViT-32/14 | 630 M | I-JEPA | ImageNet-1k | 1.3 M | 91.7 | 99.2 |
| MIM-JEPA pretrained SCOTT enabled ViTs (ours) | | | | | | |
| SCOTT-7/16 | 14 M | MIM-JEPA | Pets-37 | 7349 | 81.7 | 97.3 |
| SCOTT-7/16* | 14 M | MIM-JEPA | Pets-37 | 7349 | 88.0 | 99.0 |
| SCOTT-12/16 | 22 M | MIM-JEPA | Pets-37 | 7349 | 86.2 | 98.5 |
| SCOTT-12/16* | 22 M | MIM-JEPA | Pets-37 | 7349 | 90.7 | 99.4 |
| **ImageNet-100** | | | | | | |
| Fine-tuned ViTs from supervised pretraining (SL) | | | | | | |
| SparseSwin | 17 M | SL | ImageNet-1k | 1.3 M | 86.9 | - |
| Self-supervised learning pretrained ViTs | | | | | | |
| ViT-12/14 + reg | 22 M | DinoV2 | LVD-142M | 142.0 M | 89.1 | 98.9 |
| ViT-32/14 | 630 M | I-JEPA | ImageNet-1k | 1.3 M | 88.7 | 98.6 |
| MIM-JEPA pretrained SCOTT enabled ViTs (ours) | | | | | | |
| SCOTT-7/16 | 14 M | MIM-JEPA | ImageNet-100 | 135 K | 81.1 | 96.0 |
| SCOTT-12/16 | 22 M | MIM-JEPA | ImageNet-100 | 135 K | 84.9 | 97.5 |

(Assran et al., 2023). Remarkably, our method achieves competitive performance while training smaller models and pretraining exclusively on the target dataset, which is several orders of magnitude smaller and less heterogeneous than those used for pretraining both DINOv2 and I-JEPA. For example, on Pets-37, I-JEPA achieves a Top-1 accuracy of 91.7% with a ViT-32/14 model of 630 M parameters pretrained on ImageNet-1K (1.2 M images). In contrast, our SCOTT-12/16 (22 M parameters) achieves 90.7% top-1 accuracy while pretraining only on 7349 unlabeled images from the target dataset. Similarly, on ImageNet-100, DinoV2 attains a Top-5 accuracy of 98.9% after

pretraining on the LVD-142M dataset cromprising 142 million images, whereas our method reaches a comparable Top-5 97.5% while pretraining on only 135,000 images.

These examples illustrate that our approach achieves near state-of-the-art performance with a fraction of the data and computational resources required by existing methods. In fact, fine-tuning MIM-JEPA pretrained SCOTT models might yield even better results; however, since achieving absolute state-of-the-art performance is not the main goal of our work, we leave this exploration for future reasearch. This section demonstrates the efficiency and practicality of our method in settings where large-scale data and computational resources are not available, highlighting its potential impact across a wide range of applications. Moreover, while our method is designed to succeed on small-scale environments, the results in Table 1 suggest that it has the potential to scale well as resources increase along three axes: (i) dataset size, (ii) model size, and (iii) training time –a desirable property shared with the standard ViT.

In contrast to most generative SSL frameworks that typically require fine-tuning all model parameters, our learning framework produces robust off-the-shelf features that enable the training of simple classifiers on top. This property of discriminative SSL (Caron et al., 2021; Oquab et al., 2024) is achieved in our setup without complex image augmentations to introduce view-invariant biases.

## 5 BUILDING INTUITIONS WITH ABLATIONS

We conduct ablation studies to better understand the contributions of each component proposed: SCOTT enabled ViT and MIM-JEPA. We run ablations on 300 epochs, which yields consistent results with our best training of 1200 epochs. For all experiments in this section we keep the same pretraining recipe of Section 4.2, but remove the component in study. Specifically, we use the SCOTT-12/16 variant since its size is comparable to ViT-S, a standard ViT configuration in literature. Moreover, we select the Flowers-102 dataset to run the ablations for several reasons: (i) there are only 8189 images for MIM-JEPA self-supervised pretraining and roughly 20 labeled images per class for supervised learning. (ii) there are 102 flower classes to classify with very high intra-class similarity. (iii) they are all high-resolution images. A summary of ablations is reported in Table 2.

**SCOTT Tokenizer without MIM-JEPA pretraining**. In Table 2, we quantify the performance improvement achieved by using MIM-JEPA pretraining for learning visual representations versus supervised training from random initialization. For MIM-JEPA pretrained SCOTT models, the weights are frozen after the self-supervised learning stage, and only a lightweight classifier is trained on top. In contrast, supervised end-to-end training of the entire SCOTT model yields the poorest performance, with an 18.02-point lower top-1 accuracy. These results are particularly relevant in fields where annotated data is scarce and expensive, yet a bigger unlabeled dataset is available.

**MIM-JEPA pretraining without SCOTT Tokenizer**. In Table 2, to assess the importance of the SCOTT Tokenizer, we performed an ablation where MIM-JEPA pretraining used the standard patch embedding tokenization in ViT instead. Notably, as shown in Table 10, SCOTT-7/16 (13.6 M parameters) slightly outperforms ViT-12/16 (21.5 M parameters) while having nearly half the parameters. This characteristic is crucial for fields like robotics and embedded systems, where computational resources are more restrictive.

Table 2: Ablation studies for SCOTT models and MIM-JEPA pretraining on image classification. The first row corresponds to our proposed method, subsequent rows ablate different components.

| Models | Flowers-102 | |
|---|---|---|
| | Top-1 | Top-5 |
| SCOTT-12/16 and MIM-JEPA pretraining (300 Epochs). *(ours)* | 97.15 | 99.15 |
| - No MIM-JEPA & No SCOTT (i.e., ViT-12/16 supervised learning) | 71.08 | 87.52 |
| - No MIM-JEPA pretraining (i.e., SCOTT-12/16 supervised learning) | 79.13 | 91.96 |
| - No SCOTT (i.e., Patch and Embed Tokenization, ViT-12/16) | 95.25 | 99.07 |
| - No color augmentations | 95.86 | 98.82 |
| - Random masking (0.6 mask ratio) | 92.06 | 97.99 |

**Image augmentations**. Turning off color image augmentations results in less than a 2-point performance drop, suggesting that augmentations may not be necessary when pretraining SCOTT models

within MIM-JEPA. This is particularly relevant for fields like x-ray imaging or modalities like audio, where image-specific augmentations are not feasible. Further ablations are reported in Appendix F.

# 6 QUALITATIVE RESULTS

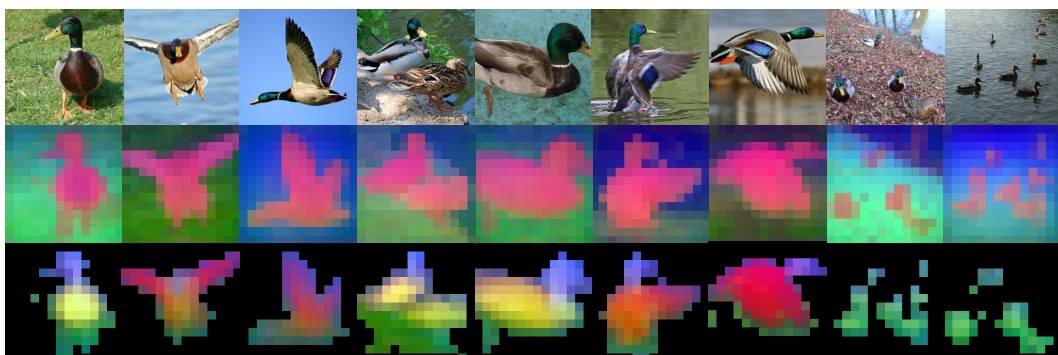

Figure 3: **Visualization of the first PCA components**. We compute a PCA between the patches from all images in the first row. A semantic class segmentation emerges in pink, the background is removed by thresholding the first component. A second PCA among remaining object's patches reveals different objects parts: the head in purple, the torso in yellow or the wings in red. Similar to Figure 1 (c), the two rightmost columns segment several ducks, potentially enabling object counting.

**PCA of patch features.** We conduct a principal component analysis (PCA) on the patch features produced by our model and present the results in Figure 3 and Figure 1. To enhance visualization, we map the first three principal components to RGB color channels. Notably, different colors correspond to different semantic "objects" or "parts" that consistently match across images of the same *family*. This emerging property -despite our model not being specifically trained to parse object parts- was previously reported in DinoV2; however, our method achieves this without relying on complex view-invariant image augmentations nor having a class token. Moreover, by thresholding the first principal component to retain only the positive values, we effectively segment the main object (foreground) from the background. By further applying a second PCA on the remaining patches, we can further separate different semantic "parts" of the main object, see Figures 1 and 3.

# 7 CONCLUSION

Effective representation learning in computer vision has traditionally required large-scale datasets and vast computational resources. In this work, we demonstrate that robust off-the-shelf representations can be learned with limited data, compute, and model sizes by integrating a Sparse Convolutional Tokenizer into Transformer architectures. SCOTT introduces CNN-like inductive biases while maintaining compatibility with masked image modeling objectives, enabling our MIM-JEPA self-supervised pretraining. Our experiments show that frozen SCOTT models pretrained with MIM-JEPA allow simple classifiers to significantly outperform fully supervised methods and achieve competitive results with state-of-the-art approaches, while using only the small-scale target datasets and not heavily relying on complex image augmentions. This is particularly relevant to a long tail of computer vision applications beyond natural images, where data and computational resources are constrained. Future work will explore fine-tuning techniques, dense prediction tasks such as image segmentation, and the application to domain-specific data like medical imaging. Continued research in escaping the big data paradigm will enhance accessibility and impact across diverse fields.

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

# A ARCHITECTURE

## A.1 SPARSE CONVOLUTIONAL TOKENIZER FOR TRANSFORMERS (SCOTT) ARCHITECTURE

Table 3: Architecture of the Sparse Convolutional Tokenizer for Transformers (SCOTT)

| Layer | Type | # in | # out | Kernel size | Stride | Padding |
|-------|------|------|-------|-------------|--------|---------|
| 1 | Sparse Convolution 2D | 3 | 64 | 7 | 2 | 3 |
| 2 | ReLU | - | - | - | - | - |
| 3 | Sparse MaxBlurPool 2D | - | - | 3 | 2 | - |
| 4 | Sparse Convolution 2D | 64 | 384 | 7 | 2 | 3 |
| 5 | ReLU | - | - | - | - | - |
| 6 | Sparse MaxBlurPool 2D | - | - | 3 | 2 | - |

## A.2 TRANSFORMER BACKBONE

Table 4: SCOTT Transformer backbone variants

| Model | Emb. Dim. | Pos. Emb. | # Blocks | # Heads | FFN | # Params |
|-------|-----------|-----------|----------|---------|-----|----------|
| SCOTT-7/16 | 384 | Fixed | 7 | 4 | SwiGLU | 13.6 M |
| SCOTT-12/16 | 384 | Fixed | 12 | 6 | SwiGLU | 22.4 M |

# B DATASETS

Table 5: Description of the datasets used in Section 4.

| Dataset | # Classes | Train size | Test size | Magnitude |
|---------|-----------|------------|-----------|-----------|
| Flowers-102 | 102 | 2040 | 6149 | $10^1$ |
| Pets-37 | 37 | 3680 | 3669 | $10^2$ |
| ImageNet-100 | 100 | 130000 | 5000 | $10^3$ |

- **Oxford Flowers-102** (Nilsback & Zisserman, 2008) The task consists in classifying among images of flowers present in the UK (102 classes, with between 40 and 248 images per class) with a total of 2040 images for training (1020 as validation split) and 6149 for evaluation. Each image dimension has at least 500 pixels.

- **Oxford IIIT Pets-37** (Parkhi et al., 2012) The task consists in classifying images of dog and cat breeds (37 classes, with around 200 pictures each). The domain-specific features challenges models to differentiate between breeds that may be visually similar. There are 3680 images for training and 3669 for testing.

- **ImageNet-100** (Deng et al., 2009) The task consists in classifying images of 100 different classes of animals present in the well-studied ImageNet dataset. There are 130000 images for training (with roughly 1300 images per class) and 5000 images for testing.

## C    IMAGE AUGMENTATIONS

During self-supervised training, MIM-JEPA uses the following image augmentations to generate different views while preserving content location:

- Random cropping: a random patch from the original image is selected with an area uniformly sampled between 0.2 and 1.0, and an aspect ratio between 3/4 and 4/3. Once cropped, the patch is resized using bicubic interpolation to the target size 224×224.
- 50% chance of horizontal flip.
- Color jittering: random uniformly change the brightness (0.4), contrast (0.4), saturation (0.2), hue (0.1), with a probability of 0.8.
- Grayscale conversion with a probability of 0.1.
- Gaussian blurring: with a probability of 0.3 for a 224x224 image, apply a square Gaussian kernel of 9x9 and a standard deviation uniformly sampled between 0.1 and 2.

In the default pretraining strategy, each image view is generated through a different augmentation pipeline. First random cropping and horizontal flipping take place, then the order in which color jitter, grayscale and gaussian blurring augmentations are applied is uniformly sampled before applying the pipeline. Once that augmentation pipeline is applied, color channels are normalized by subtracting the average color and dividing by the standard deviation, computed on ImageNet.

## D    MIM-JEPA PRETRAINING HYPERPARAMETERS

Table 6: MIM-JEPA pretraining hyperparameters

| Parameter | Value |
|---|---|
| Predictor # Blocks | 3 |
| Masking | Blockwise |
| Mask ratio | 0.6 |
| Batch size | 128 |
| Optimizer | AdamW |
| # Epochs | 300 |
| Learning rate start | 0.000001 |
| Learning rate peak | 0.0005 |
| Learning rate final | 0.00001 |
| Learning rate flat (%) | 72 |
| # Linear warmup epochs | 40 |
| Learning rate decay Schedule | Cosine |
| Weight decay start | 0.04 |
| Weight decay end | 0.4 |
| Weight decay Schedule | Linear |
| EMA start | 0.996 |
| EMA end | 1.0 |
| EMA Schedule | Linear |

SCOTT models that are pretrained for longer, i.e., 1200 epochs, also warmup for longer, i.e., 60 epochs. The rest of hyperparameters is kept the same as in Table 6.

## E    EVALUATION

### E.1    EVALUATION PROTOCOLS

Given an input image, the SCOTT model pretrained using MIM-JEPA outputs a sequence of features $s = \{s_i\}_{i=1}^{N}$, where $s_i$ is the encoded semantic representation associated with the $i^{th}$ image patch. A feature pooling operation is applied to $s$ to generate a single feature vector, which is then fed into a linear classifier for downstream supervised tasks. Following literature, we report results obtained

with two different pooling strategies: a linear operation (average pooling) and a non-linear operation (attentive pooling).

**Linear Probing**. To pool the sequence of features $s = \{s_i\}_{i=1}^N$ into a single vector, a simple linear operation (average pooling) is applied, followed by a LayerNorm. The resulting feature vector is fed into a linear classifier.

**Attentive Probing** (Bardes et al., 2024). To pool the sequence of features $s = \{s_i\}_{i=1}^N$ into a single vector, a lightweigth non-linear cross-attention block with a learnable query token is learnt. The output of the cross-attention block is added back to the query token through a residual connection and fed into a SwiGLU layer, followed by a LayerNorm. The resulting feature vector is fed into a linear classifier.

### E.2 EVALUATION DETAILS

Details regarding numbers reported in Table 1. For fair comparisons, unless stated otherwise all methods share the same image augmentations and hyperparameters as presented in Table 6:

- Supervised ViTs and SCOTT variants are trained for 300 epochs.
- Fine-tuned ViTs are extracted from (Steiner et al., 2021).
- DinoV2 uses a linear-probe on CLS token. Pretrained weights are publicly available. The ViT-12/14 is distilled from a ViT-g/14 (1,100 M parameters).
- I-JEPA uses an attentive-probe on patch tokens. Only pretrained weights for big model sizes (ViT-32/14) are publicly available.
- All self-supervised methods reported, i.e., DinoV2, I-JEPA, MIM-JEPA, are probed on best result after 100 epochs on the target dataset.
- SparseSwim result is from (Pinasthika et al., 2024).

## F ABLATIONS

**Masking strategy**. In Table 7 we compare different masking strategies. Blockwise masking is our default strategy introduced in Section 3.2. In random masking the target is a set of patches uniformly sampled from the encoded image representation . For both masking strategies, the context image is the complement of the masked target set, ensuring that there are no overlapping patches between the context and target blocks. Consistent with prior works, we find that MIM-JEPA benefits more from blockwise masking than from random masking. The intuition is that blockwise masking strikes a good balance in generating target blocks with relative semantic meaning while producing context blocks that are informative of the missing information. Additionally, higher masking ratios also improve performance.

Table 7: Ablating masking strategy. Attentive and linear evaluation on Flowers-102 Dataset using the train split (2040 labeled samples) after MIM-JEPA pretraining of a SCOTT-12/16 enabled ViT for 300 epochs. Blockwise masking achieves superior performance in both attentive and linear evaluation. In addition, a higher masking ratio leads to better performance overall.

| M strategy | M ratio | Top-1 Attentive | Top-1 Linear | Top-5 Attentive | Top5 Linear |
|---|---|---|---|---|---|
| Random | 0.4 | 90.64 | 81.57 | 97.64 | 95.00 |
| Random | 0.6 | 92.04 | 84.46 | 97.99 | 95.91 |
| Blockwise | 0.4 | 95.85 | 92.66 | 98.86 | 98.38 |
| Blockwise | 0.6 | **97.15** | **94.81** | **99.15** | **98.78** |

**Image augmentation strategy.** In the default MIM-JEPA pretraining strategy, we generate two (different) views of a given crop with a certain probability by slightly modifying only the color properties; thereby, preserving equivalent spatial content. We ablate the performance of this strategy versus applying the same color augmentation to both views (same) and to disabling color augmentations entirely (none). As shown in Table 8, (different) view augmentation strategy achieves best

performance. However, it is noteworthy that the performance gap compared to using no augmentations (none) is less than 2 percentage points. This suggests that augmentations may not be necessary when pretraining SCOTT models within a MIM-JEPA framework. The intuition is that the JEPA objective of predicting in abstract representation space potentially mitigates the reliance on unnecessary pixel-level details. This is particularly relevant to fields (e.g., x-ray imaging) and modalities (e.g., audio) where image-specific augmentations are not feasible.

Table 8: Performance Comparison of Image Augmentation Strategies. The "different" view augmentation strategy achieves the highest performance across metrics. However, the performance gap compared to using no augmentations ("none") is less than 2 percentage points, suggesting that augmentations may not be necessary when pretraining SCOTT models within a MIM-JEPA framework.

| Augmenation Strategy | Top-1 Attentive | Top-1 Linear | Top-5 Attentive | Top5 Linear |
|---|---|---|---|---|
| none | 95.86 | 92.60 | 98.82 | 97.82 |
| same | 96.76 | 94.29 | 99.12 | 98.56 |
| different | **97.15** | **94.81** | **99.15** | **98.78** |

Table 9: Performance comparison of SCOTT models with and without MIM-JEPA pretraining. The results demonstrate that MIM-JEPA pretraining significantly improves top-1 accuracy by 18 percentage points compared to supervised training from scratch, even when only a lightweight classifier is trained on top of frozen pretrained weights.

| Pretraining strategy | Ptretraining data size | Supervised Training size | Top-1 Attentive | Top-1 Linear | Top-5 Attentive | Top-5 Linear |
|---|---|---|---|---|---|---|
| None | - | Train split (2040) | 79.13 | 78.54 | 91.96 | 91.85 |
| MIM-JEPA | Train split (2040) | Train split (2040) | 80.69 | 66.92 | 93.83 | 87.03 |
| MIM-JEPA (ours) | Train + Test (8189) | Train split (2040) | 97.15 | 94.81 | 99.15 | 98.78 |

Table 10: Performance comparison of MIM-JEPA pretraining with and without SCOTT Tokenizer. This table illustrates the importance of the SCOTT Tokenizer by comparing models where MIM-JEPA pretraining uses the standard patch embedding in ViT instead of the SCOTT Tokenizer. Notably, SCOTT-7/16 (13.6 M parameters) slightly outperforms ViT-12/16 (21.5 M parameters) despite having nearly half the parameters.

| Model | # Params | Top-1 Attentive | Top-1 Linear | Top-5 Attentive | Top-5 Linear |
|---|---|---|---|---|---|
| ViT-7/16 | 12.7 M | 93.54 | 89.81 | 98.69 | 97.91 |
| ViT-12/16 | 21.5 M | 95.25 | 92.82 | 98.78 | 98.40 |
| SCOTT-7/16 | 13.6 M | 95.64 | 92.70 | 99.07 | 98.19 |
| SCOTT-12/16 | 22.4 M | **97.15** | **94.81** | **99.15** | **98.78** |

## G  SCALABILITY ASSESSMENT

High scalability is one of the primary advantages of the standard ViT. In this section, we aim to assess whether this property persists when replacing its patch and embed tokenizer by a SCOTT tokenizer and pretraining within the MIM-JEPA framework. Specifically, we report Top-1 and Top-5 Attentive Probing metrics on Flowers-102 as we scale a SCOTT model along three different axes: (i) pretraining dataset size, (ii) model size, and (iii) pretraining time. While our method is designed to perform well with scarce resources, results in Table 11 suggest that not only do SCOTT and MIM-JEPA scale favorably, but they also outperform the standard ViT architecture when computational resources are limited.

**Scaling data size**. MIM-JEPA pretraining exhibits improved performance when pretrained with larger datasets. This outcome aligns with expectations, as additional data enables the model to learn more general and abstract representations that effectively distinguish between different classes.

**Scaling model size**. MIM-JEPA pretraining benefits from larger encoder sizes when pretraining on Flowers-102. We increase model sizes by adding more transformer encoder blocks, while keeping the SCOTT tokenizer intact. The predictor network is also kept constant among the different setups.

**Scaling pre-training time**. A longer MIM-JEPA pretraining time helps the model to produce slightly better image representations.

Table 11: Scalability assessment of SCOTT models pretrained on MIM-JEPA.

| Scalability assessment | Flowers-102 | |
| --- | --- | --- |
| | Top-1 | Top-5 |
| Pretraining dataset size | | |
| 1020, i.e. train split (12%). | 74.25 | 90.82 |
| 2040, i.e. train+val (25%) | 80.69 | 93.83 |
| 6149, i.e. test split (75%) | 91.88 | 97.70 |
| 8189, i.e. train+val+test (100%) | 97.15 | 99.15 |
| Model size (# parameters) | | |
| SCOTT-3/16 (6.5 M) | 93.64 | 98.60 |
| SCOTT-7/16 (13.6 M) | 95.64 | 99.07 |
| SCOTT-9/16 (17.1 M) | 96.50 | 99.25 |
| SCOTT-12/16 (22.4 M) | 97.15 | 99.15 |
| Total pretraining time | | |
| 300 epochs | 97.15 | 99.15 |
| 600 epochs | 97.59 | 99.21 |
| 1200 epochs | 97.73 | 99.21 |

## H CODE IMPLEMENTATION

To facilitate reproducibility of our work, we will release the full code implementation, including configuration files and pretrained models, in the near future.

