# OpenReview forum: "Escaping the Big Data Paradigm in Self-Supervised Representation Learning"
_ICLR.cc/2025/Conference — Submitted to ICLR 2025_

### Official Review · Reviewer_6LsZ · 2024-10-31

**Soundness:** 3
**Presentation:** 3
**Contribution:** 3
**Rating:** 5
**Confidence:** 4

**Summary:**

This work demonstrates that robust off-the-shelf representations can be learned with limited data, compute, and model sizes by integrating a Sparse Convolutional Tokenizer into Transformer architectures. The authors introduce CNN-like inductive biases while maintaining compatibility with masked image modeling objectives, enabling the self-supervised pretraining for Masked Image Modeling. To show the advantages of the paper, the authors provide extensive comparisons with other baseline methods on several downstream tasks. The authors also conducted an ablation study to show the effectiveness empirically.

**Strengths:**

+ The paper is well-organized and easy to read.
+ The paper proposes a Joint-Embedding Predictive Architecture for the Masked Image Modeling task, enabling self-supervised pre-training on a much smaller dataset.
+ This paper provides strong performance across all the tasks and architecture in a self-supervised learning setting.

**Weaknesses:**

- The difference between the proposed Sparse Convolutional Tokenizer for Transformers (SCOTT) and SparK is not obvious, it looks more like a simple leverage of previous work. The authors need to claim more of the difference with previous work.
- Experimental results with different settings are not very comparable. As for the model size, pre-training datasets, pre-training method setting are all different from the method proposed in the paper. Although the author claims that achieving absolute performance is not the main goal, the results are supposed to be comparable. For example, experiments can be added to utilize Dino/I-JEPA or other pre-training paradigms to train on the small dataset and compare it with the proposed method.

**Questions:**

Please refer to the weakness part.

---

> ### Author Response · Authors · 2024-11-15
> **Response to Reviewer 6LsZ**
>
> We would like to thank Reviewer 6LsZ for their valuable review. We would like to address the weaknesses statements in order to reinforce the understanding on the importance and novelty of our work:
>
> __W1 Distinction between SCOTT and SparK.__ As noted in the paper, SparK introduced the use of sparse convolutions to enable CNN architectures for masked image modeling (MIM) tasks. However, SparK and our work differ significantly in both architectural design and learning framework:
>
> - __Architecture:__ SparK proposes a fully convolutional encoder-decoder architecture reminiscent of UNet, which is far from SCOTT, a shallow CNN tokenizer to replace the ViT ineffiecient patch-and-embed. We leverage on SparK contributions to design sparse layers of the SCOTT tokenizer. While introducing SCOTT might seem obvious to improve efficiency of ViTs this should not be taken for granted, to the best of our knowledge there is no prior work to propose a CNN-like tokenizer for ViTs that is compatible with MIM tasks, nor to demonstrate its effectiveness. This is indeed our contribution; we are the __first__ to propose to replace the patch and embed strategy of ViTs by a shallow convolutional tokenizer that is compatible with MIM tasks to enable efficient training and prove it.
>
> - __Learning Framework:__ Spark trains the UNet in a BERT style generative framework, where the task is to predict the masked input signal. In contrast, we propose to train a ViT enabled SCOTT model in a MIM-JEPA framework, which is not generative and targets are generated by a momentum-based target network in abstract representation space, where the noise present in the input signal is potentially eliminated.
>
> In summary, while SparK contributed foundational ideas around sparse convolutions, SCOTT represents a distinct approach designed to integrate sparse convolutions within ViTs in an SSL context, making it uniquely suited for efficient training on small, fine-grained datasets.
>
> __W2 Concerns on comparability of experimental setting.__ The primary goal of our work is to propose a method that enables __effective representation learning on small-scale, fine-grained datasets without requiring extensive data or computational resources.__ With this objective, our experimental design focuses on assessing our proposed contributions -architecture (SCOTT) and learning framework (MIM-JEPA)- to determine their capability to perform competitively with limited data and compute, compared to state-of-the-art methods that depend on large-scale resources for pretraining.
>
> Given this aim, we selected baselines that represent state-of-the-art (SOTA) performance across different learning paradigms, as we believe this context provides a meaningful basis for evaluating our results. Specifically, we included:
> - __Fine-Tuned Vision Transformers (ViTs) from Supervised Pretraining:__ These models represent the top performance achieved in fully supervised setting (i.e., ViT, and SparseSwin).
> - __Self-Supervised Learning (SSL) Pretrained ViTs:__ We also include SSL-pretrained ViTs (e.g., DinoV2, I-JEPA) to establish a baseline performance of SOTA self-supervised models pre-trained on large datasets.
>
> We would like to emphasize that our method operates at a clear disadvantage relative to these baselines, as SCOTT variants (i.e., SCOTT-7) use smaller model sizes and pre-train on much smaller datasets. Despite these constraints, SCOTT achieves performance comparable to larger-scale methods, which we believe highlights the strength of our approach and reinforces the contributions of our work, rather than detracting from the comparability of the experiments.
>
> __Reviewer 6LsZ : “For example, experiments can be added to utilize Dino/I-JEPA or other pre-training paradigms to train on the small dataset and compare it with the proposed method.”__
>
> We note that directly pre-training Dino or I-JEPA on small datasets could offer valuable insights. However, such experiments are computationally intensive and may not reflect optimal training conditions for the standard ViT models which are data thirsty. For instance, Dino contrastive learning objective relies on the forward pass of many different local and global views, resulting in substantial GPU memory consumption (which is beyond our current available resources :-( ).
>
> We thank Reviewer 6LsZ for their valuable feedback and the opportunity to clarify our work’s distinctions and objectives. We believe our method’s ability to achieve competitive performance on small-scale datasets without large-scale resources is a meaningful advancement in accessible self-supervised learning. We hope that our responses address the reviewer’s concerns and reinforce the novelty and impact of our contributions, and we respectfully encourage the reviewer to consider reevaluating our work in light of these clarifications. We are looking forward to the reviewer’s response and appreciate their time and consideration. We look forward to the Reviewer's response.

---

### Official Review · Reviewer_BkxW · 2024-11-02

**Soundness:** 3
**Presentation:** 3
**Contribution:** 2
**Rating:** 5
**Confidence:** 4

**Summary:**

The paper introduces two advancements in self-supervised learning from images with limited data SCOTT (Sparse Convolutional Tokenizer for Transformers) and MIM-JEPA (Masked Image Modeling with Joint-Embedding Predictive Architecture). SCOTT infuses convolutional biases into ViTs, enhancing their effectiveness in data-constrained environments, while MIM-JEPA optimizes the representation learning in a latent space. This dual approach reduces the dependency on large-scale datasets, enabling effective training on datasets like Oxford Flowers-102, Oxford IIIT Pets-37, and ImageNet-100.

**Strengths:**

1. The integration of convolutional biases through SCOTT and the focus on semantic feature extraction via MIM-JEPA can shift away from the reliance on extensive pre-training datasets.

2. The proposed methods outperform fully supervised methods and achieve results competitive with state-of-the-art models pre-trained on much larger datasets.

**Weaknesses:**

1. The authors claim that the datasets used are high-resolution; however, I believe these datasets should not be considered high resolution.  (Of course, compared to low-resolution CIFAR and MNIST, there are). I suggest that the authors also include results from higher, domain-specific resolution datasets, as well as from low-resolution datasets, to provide a more comprehensive analysis of performance variations across different resolutions.

2. The methodology appears to be primarily limited to classification tasks. Although the authors mention that future work will extend to segmentation, it would be beneficial if they could discuss the potential applicability of their methods to segmentation tasks more explicitly.

3. Fine-tuning on pre-trained general models might still be the best way to train domain-specific images, offering less training time and potentially better performance. The authors should consider comparing their approach directly to traditional fine-tuning methods to substantiate their claims and highlight any genuine advantages or limitations.

**Questions:**

1. What specific advantages do convolutional biases offer over other techniques designed to improve data efficiency in vision models, such as attention augmentation or advanced data augmentation techniques?


2. Can the authors provide preliminary insights on how their approach might be adapted for segmentation?

---

> ### Author Response · Authors · 2024-11-15
> **Response to Reviewer BkxW**
>
> We thank Reviewer BkxW for the helpful comments and valuable review, especially for recognizing that our key contributions --SCOTT and MIM-JEPA-- can shift away the reliance on extensive pre-training datasets, which is the main goal of our work. We appreciate the reviewer’s acknowledgment that our method outperforms fully supervised approaches and achieves results competitive with state-of-the-art models pre-trained on much larger datasets.
>
> However, given these positive insights, we are somewhat unclear about the basis for the initial score of 3 (“reject, not good enough”). We believe this score may be influenced by the noted weaknesses and questions, which we feel may stem from a misunderstanding of our core contributions. We proceed to address these points in detail below:
>
> __W1. On dataset resolution terminology.__ We agree that the term “high-resolution” can be somewhat subjective and may lead to different interpretations depending on the audience's background. In  Image classification literature there is often crisp distinction between “high” and “low” resolution datasets, where “high-resolution” is often used to distinguish datasets like Flowers-102 and Pets-37 from low-resolution datasets such as CIFAR or MNIST. Our intent was simply to clarify that our datasets contain images of higher resolution compared to these low-resolution benchmarks, which are not within the scope of our work.
>
> We recognize that in other fields, such as image generation, “high-resolution” often refers to even larger image sizes. We are open to adjusting the terminology if Reviewer BkxW or others have suggestions for a more precise description that would prevent potential misunderstandings.
>
> __W2 and Q2. Extending to dense prediction tasks.__ As stated in the paper, “We focus on classification because many industrial and medical applications rely on classification (e.g., disease or defect detection).” This aligns with our primary objective of addressing needs in domains where classification is central, and data is often limited.
> Regarding dense prediction tasks, such as segmentation, a key advantage of introducing a convolutional tokenizer like SCOTT is the flexibility to build a UNet-like decoder. With SCOTT, progressive downsampling feature maps can be used as skip connections to guide the upsampling process in the decoder, making SCOTT well-suited for dense prediction. This flexibility is not possible with the patch-and-embed tokenization strategy of standard ViTs, which lacks these spatial hierarchies.
> Although we briefly mentioned dense prediction as a future direction, we refrained from expanding on this potential benefit in the current paper, as there were no experiments conducted to fully support these foreseeable applications. We plan to explore this in future work to validate SCOTT’s applicability to dense prediction tasks.
>
> __W3. Comparing to traditional fine-tuning methods:__ In terms of comparing our approach to traditional finetuning methods, we direct Reviewer BkxW to Table 1 in our original manuscript, where we report results of ViT-12/16 pretrained on ImageNet (1k and 21k) and finetuned on the target datasets. We would like to take the opportunity to highlight that our method SCOTT + MIM-JEPA achieves comparable results while using way less resources across different axes, this is the main contribution of our work.
>
> While fine-tuning pretrained models may remain the preferred option in the natural images domain, our objective is to demonstrate that SCOTT+MIM-JEPA can enable competitive training on domain-specific datasets with limited resources. This capability is particularly relevant for a wide range of computer vision tasks in resource-constrained fields (e.g., medical imaging, robotics) where pretraining on massive datasets is not feasible.
>
> As part of our future work, we plan to extend our research beyond natural images, focusing on domains where larger, pretrained models currently dominate. We believe that our approach has the potential to serve as a valuable alternative in these areas, and we do not claim otherwise.

---

> ### Author Response · Authors · 2024-11-15
> **Response Continuation to Reviewer BkxW**
>
> __Q1. Specific advantages of CNN biases over Augmentation techniques.__ Introducing convolutional biases is a more fundamental approach to improving data efficiency compared to augmentation techniques. Augmentation techniques expand the dataset by applying transformations, but they do not enhance the model’s inherent capacity to generalize from the data structure itself. In contrast, -extracted from the original manuscript- “CNNs, inspired by the hierarchical processing of the mammalian visual cortex (Hubel & Wiesel, 1959; Fukushima, 1988), provide important priors for learning spatial relationships in visual data.”
>
> Moreover, designing effective augmentation strategies is often domain-specific and requires expert knowledge, especially outside of the natural images domain. Convolutional biases, however, are inherently compatible with the spatial structure of image data, making them broadly applicable and effective across various domains.
>
> While convolutional biases provide this general advantage, they remain fully compatible with augmentation techniques, and the two approaches can be complementary in cases where further data transformations may enhance performance.
>
> We appreciate Reviewer BkxW's thoughtful feedback and valuable recognition of our contributions. We hope that our responses clarify key aspects of our approach, particularly the unique advantages of SCOTT and MIM-JEPA in advancing data-efficient learning beyond reliance on extensive pre-training datasets. Given these points, we respectfully encourage the reviewer to consider updating their initial evaluation, as we believe these clarifications highlight the relevance and potential impact of our work. Thank you once again for your time and constructive insights. We look forward to the Reviewer BkxW's response.

---

> > ### Comment · Reviewer_BkxW · 2024-12-02
> >
> > Thank the authors for their response and clarification. I raise my score to borderline.

---

### Official Review · Reviewer_7fBC · 2024-11-03

**Soundness:** 1
**Presentation:** 2
**Contribution:** 1
**Rating:** 3
**Confidence:** 4

**Summary:**

This paper proposes Sparse Convolutional Tokenizer for Transformers (SCOTT) which is a tokenization architecture that injects convolutional inductive biases into Vision Transformers. The purpose it enable small-scale data training while compatible with MIM tasks. The author of the paper also proposes Joint-Embedding Predictive Architecture within a MIM framework. The experiments on small-scale dataset shows promising results.

**Strengths:**

1. This paper is easy to follow and has clean organization.

2. This topic is promising since training Transformer-based vision model is very data thirsty.

**Weaknesses:**

1. The comparison experiments in the paper is weak since there are tons of conv+ViT baselines. This paper, however, only compare to a few, also the related works missed many related references. Therefore, the paper’s experiments is not quite convincing.

2. The motivation is clear but this paper lacks the analysis of related works. What kind of problems are there for similar design? Why the proposed method is better? Why choosing such design (e.g., MIM-JEPA)? The overall elaboration is not quite self-sufficient.

3. The experiments are not sufficient to demonstrate the effectiveness of the method. The settings and comparison is too simple, and very limited ablations are conducted.

**Questions:**

Please refer to weaknesses.

---

> ### Author Response · Authors · 2024-11-15
> **Response to Reviewer 7fBC**
>
> We appreciate Reviewer 7fBC for recognizing the importance of this work, describing it as “promising since training Transformer-based vision model is very data thirsty”. However, we noted some potentially contradictory points in the review: the summary mentions that "the experiments on small-scale datasets show promising results," yet the weaknesses section states that “the comparison experiments in the paper are weak” and “the experiments are not sufficient to demonstrate the effectiveness of the method.”
> We appreciate the opportunity to clarify and expand on our experimental section to address these points. In this response, we will outline our approach and address specific questions from Reviewer 7fBC.
>
> To recall, the primary goal of our work is to propose a method that enables __effective representation learning on small-scale, fine-grained datasets without requiring extensive data or computational resources.__ With this goal, our experimental design focuses on assessing our proposed contributions (architecture: SCOTT and learning framework: MIM-JEPA) capability to perform competitively with limited data and compute, compared to state-of-the-art methods that depend on large-scale pretraining.
>
> __W1. Comparison experiments to Conv+ViT baselines.__ We would like to clarify that our primary goal is not to propose a new conv+ViT architecture achieving state-of-the-art performance in supervised learning (SL). Instead, our focus is to: __“enable effective representation learning on small-scale, fine-grained datasets without requiring extensive data or computational resources”.__
>
> To achieve this, we propose SCOTT—a novel approach that integrates a convolutional tokenizer with Vision Transformers (ViTs) in a way that is compatible with masked image modeling (MIM) within a self-supervised learning (SSL) framework. To the best of our knowledge, this is the first work to incorporate a convolutional tokenizer with ViTs specifically for MIM-based SSL, making SCOTT a unique contribution within the SSL context.
>
> Given this novelty and focus, we selected baselines that represent state-of-the-art (SOTA) performance across different learning paradigms, as we believe this provides a more meaningful comparison for our results, regardless of whether these baselines employ conv+ViT architectures.
> Specifically, we included:
>
> -	__Fine-Tuned Vision Transformers (ViTs) from Supervised Pretraining:__ These models represent the top performance achieved in fully supervised setting (i.e., ViT, and SparseSwin).
>
> -	__Self-Supervised Learning (SSL) Pretrained ViTs:__ We also include SSL-pretrained ViTs (e.g., DinoV2, I-JEPA) to show the baseline performance of SOTA self-supervised models trained on large datasets.
>
> We did not include conv+ViT supervised models in Table 1, as these underperform compared to the SOTA baselines selected. Instead, fully supervised SCOTT enabled ViT performance is reported as a representative example of a conv+ViT model trained from scratch in fully supervised conditions for the same number of epochs (300), where it outperforms standard ViTs under the same settings but underperforms any of the pretrained models.
>
> Regarding the Related Works section of the manuscript, given the page-limit constraint, we included references to conv+vit works that either achieved top results or are directly relevant to our proposed method. For instance, we noted, “Recognizing this limitation, numerous studies have previously explored incorporating convolutional priors into ViT architectures (Wu et al., 2021; Chen et al., 2021; Yuan et al., 2021; Graham et al., 2021).”.  If there are specific works that Reviewer 7fBC finds missing or particularly relevant, we would appreciate the recommendations and will revisit the Related Works section to include these in the final version.

---

> ### Author Response · Authors · 2024-11-15
> **Continuation to Previous Response To Reviewer 7fBC**
>
> __W2.__ We will proceed by citing our own manuscript to answer the different questions. In essence, our design choices are to address the fundamental difference between computer vision and natural language processing, which we demonstrate with our contributions that have been overlooked in prior works in favor of “using more data”.
>
> __Reviewer 7fBC: What kind of problems are there for similar design? Why is the proposed method better? Why choosing such design (e.g., MIM-JEPA)?__
>
> - __Convolutional tokenizer in ViT for MIM:__ “introducing a CNN tokenizer conflicts with the patch-wise masking strategy because one cannot eliminate pixel information from masked patches -to avoid trivial solutions- as ViTs do byremoving or replacing them with a mask token.”
>
> - __Generative MIM architectures:__ “Since the introduction of MIM, various methods have explored different reconstruction targets, such as raw pixels (He et al., 2022; Xie et al., 2020; 2022), or patch-level tokens via a learned tokenizer (Bao et al., 2021; Peng et al., 2022). While these approaches have been effective in scaling self-supervised learning to larger datasets, they often lead to feature representations at a low-level of semantic abstraction” … “JEPAs are conceptually close to Generative Architectures, however, the loss function is applied in embedding space, not input space”, that is, the model does not reconstruct the noise present in the input signal space and can focus on semantic meaning.
>
> “Building on these ideas, we integrate our Sparse Convolutional Tokenizer for Transformers (SCOTT) within the ViT architecture of a JEPA framework based on MIM and dubbed MIM-JEPA. This combination enables effective self-supervised learning on small-scale datasets, where traditional ViT approaches typically struggle.”
>
> __W3 Experiments and ablations.__ We would appreciate further clarification from Reviewer 7fBC regarding concerns about the experiments and ablations. As noted in the paper, __“We focus on classification because many industrial and medical applications rely on classification (e.g., disease or defect detection),”__ and the main goal of our method is to provide a viable solution for domains where resources and labeled data are limited. With this in mind, we selected Flowers-102, Pets-37 and ImageNet-100, which are particularly challenging due to their high intra-class similarity, making it harder to distinguish between categories (e.g., different types of flowers) compared to more generic categories (e.g., persons, cars, planes).
>
> In Table 1, we report results across the most relevant available learning paradigms:
> - __Fully supervised learning:__ Training ViTs and SCOTT from scratch.
> - __Transfer Learning from Supervised Pretraining:__ Fine-tuned ViTs pretrained on large datasets.
> - __Probing State-of-the-Art (SOTA) Self-Supervised Models:__ Evaluating the performance of DinoV2 and I-JEPA pretrained on large datasets.
> - __Our Method (SCOTT+MIM-JEPA):__ To assess its effectiveness in small-scale, data-limited settings.
>
> We believe these comparisons provide a comprehensive view of SCOTT+MIM-JEPA’s performance against key learning paradigms. However, if Reviewer 7fBC has specific suggestions for additional experiments, we would appreciate the feedback on what additional comparisons might further demonstrate our method’s effectiveness in these settings.
>
> __Regarding incomplete ablations (see Table 2)__, we presented several analyses to examine the impact of key components in our architecture and learning framework. Specifically:
> - No MIM-JEPA and No SCOTT, i.e., a ViT trained in supervised learning.
> - No MIM-JEPA pretraining, i.e., a ViT enabled with SCOTT tokenizer in supervised learning.
> - No SCOTT, that is a ViT with patch-and-embed but pretrained using MIM-JEPA.
> - No color augmentations.
> - Random masking instead of Blockwise masking.
>
> If there are additional ablations that Reviewer 7fBC feels would be valuable to include, we would appreciate any specific recommendations and will include them in our final version.
>
> We hope this detailed clarification offers Reviewer 7fBC an updated perspective on our work and its contributions toward advancing representation learning beyond the big data paradigm, thereby making computer vision more accessible to resource-limited domains. Given these points, we respectfully ask the reviewer to reconsider the initial evaluation of our work. We appreciate the reviewer’s time and look forward to the Reviewer's 7fBC response.

---

### Official Review · Reviewer_M9yc · 2024-11-04

**Soundness:** 2
**Presentation:** 2
**Contribution:** 2
**Rating:** 3
**Confidence:** 3

**Summary:**

This paper introduces SCOTT, a Sparse Convolutional Tokenizer designed to enhance Vision Transformers (ViTs) by incorporating convolutional inductive biases, enabling effective self-supervised learning on small datasets. SCOTT integrates with MIM-JEPA, a Joint-Embedding Predictive Architecture within a Masked Image Modeling (MIM) framework, to capture higher-level semantic features. The approach is validated on fine-grained datasets, such as Oxford Flowers-102 and Oxford IIIT Pets-37, achieving competitive results with significantly fewer data and computational resources.

**Strengths:**

1. This paper addresses an important problem: enabling model training on small-scale, unlabeled datasets, which is critical for advancing self-supervised learning in data-limited settings.

2. The authors conduct extensive experiments using multiple datasets.

**Weaknesses:**

1. The contributions of the proposed methods appear incremental compared to previous work.

2. The evaluation and comparisons with baseline and prior methods seem unfair due to differences in training setups.

3. The writing quality could be improved for clarity and readability.

Please see my comments below for further details.

**Questions:**

This paper proposes SCOTT and MIM-JEPA, two components that collaboratively enable effective model training on small-scale, unlabeled datasets. Together, they achieve promising results and open avenues for future research in resource-constrained settings. However, I have the following questions and concerns.

1. The novelty of SCOTT appears limited. Many prior works have explored injecting convolutional layers into vision transformers, as mentioned in the paper. The key challenge in combining convolutional layers with masked image modeling (MIM) is that the masked areas can diminish due to the convolutional nature. However, sparse convolution techniques, including submanifold sparse convolution, have been well-established for managing masked areas, and it seems that SCOTT directly adopts these existing techniques. Could the authors elaborate on the unique contributions of SCOTT over these previous approaches?

2. I am unclear about the novelty of MIM-JEPA compared to I-JEPA. The training pipelines for the two methods seem very similar. Could the authors provide further details to clarify the specific contributions of MIM-JEPA beyond what is already achieved by I-JEPA?

3. The evaluation methodology raises concerns about fairness:

   3.1 For the baseline of training a model from scratch with fully supervised learning, is only the final (or several) layers trained, or is the entire model fine-tuned? From the text, it appears to be the former, which would weaken this baseline and result in significantly lower accuracy compared to pre-trained methods. To properly evaluate the effectiveness of supervised learning, which is generally a strong baseline, the entire model should be trained for the same number of epochs as in pre-training (300 or 1200 epochs).

   3.2 When comparing with SSL pre-training baselines and other SSL works, the datasets used for pre-training differ, raising concerns about fairness. While prior methods are pre-trained on larger datasets like ImageNet or LVD-142M, SCOTT+MIM-JEPA is pre-trained on the target dataset itself, which is then also used for evaluation (e.g., attention or linear probing). This overlap can lead to an advantage, as the model learns features directly from the target dataset. For a fair comparison, SCOTT+MIM-JEPA should pre-train on a small, unrelated dataset. For instance, if Flowers-102 is used for evaluation, then Pets-37 or ImageNet-100 could serve as a pre-training dataset.

4. The accuracy of SCOTT+MIM-JEPA is still notably lower than models trained on large-scale data. While this is expected, the significant accuracy gap makes it difficult to consider the approach "promising," especially given the aforementioned evaluation concerns.


minor point(s):

1. The method is tailored specifically for MIM-related self-supervised learning, which limits its application scope, as it is not compatible with contrastive learning. However, given the popularity of MIM, this specialization is understandable and not a major issue.

2. The paper's writing could be improved. The paper references many prior works that inspired it, but the current organization makes it challenging to distinguish the unique contributions of this work.

3. Introducing I-JEPA in the background section would be helpful, as the current work builds directly upon it.

---

> ### Author Response · Authors · 2024-11-15
> **Response to Reviewer  M9yc**
>
> We thank Reviewer M9yc for the time and thoughtful feedback. However, some of the reviewers’ statements (e.g., “the accuracy gap between our method and large-scale models make it difficult to consider this approach promising”, “for instance, if Flowers-102 is used for evaluation, then Pets-37 or ImageNet-100 could serve for pre-training”) suggest a possible misunderstanding of the core objectives and contributions of our work. We believe these interpretations may have led to an underestimation of the novelty and impact of our approach.
>
> To clarify and reinforce the main contributions, we would like to restate the key objectives of our work as outlined in the Abstract and other sections of the manuscript:
>
> - __The primary goal of this paper is to propose a method to: “enable effective representation learning on small-scale, fine-grained datasets without requiring extensive data or computational resources”.__ Although we compare our results directly to state-of-the-art (SOTA) supervised finetuned models and SSL methods such as DinoV2 and I-JEPA (both of which are trained on large-scale data), our objective is not to outperform these large-data trained models. Rather, we aim to demonstrate that similar results can be achieved with significantly fewer resources. We hope this clarification shows that judging our method from the perspective of large-data SOTA approaches overlooks the resource constraints our method was specifically designed to address.
> - __Our key contribution is in addressing the challenge above__, which is relevant to a long tail of computer vision applications where large-scale data is either unavailable or costly to obtain. Achieving this requires non-trivial efforts across both architecture and learning framework, to effectively address the fundamental differences between NLP and CV is SSL, which we believe have been overlooked in prior work in favor of “using more data”.
> - __Learning framework:__ previous MIM frameworks, such as BEIT, iBOT which DinoV2 builds upon, or I-JEPA, to mention some, proved that masked modeling can effectively work on vision tasks by leveraging the ease with which ViTs can mask patches. However, these frameworks rely heavily on large-scale data to achieve competitive results. In contrast, our work is the first to achieve comparable results without requiring large-scale datasets, making SSL more accessible to low-data regimes.
> - __ViT Architecture:__ through previous literature in Supervised Learning (not SSL), we observed that the patch and embedding tokenization strategy in ViTs is a major contributor to its data inefficiency. While previous literature, such as CCT, have shown that a convolutional tokenizer improves data inefficiency, conventional convolutions are not compatible with MIM. To overcome this, “following pioneering work of Spark to enable BERT pre-training on CNN architectures”, we propose to introduce a shallow sparse convolutional tokenizer as a drop in replacement for the patch-and-embed in ViTs. To our knowledge, this is the first such approach for MIM-compatible convolutional tokenization in ViTs. If Reviewer M9yc is aware of prior works that introduce this idea, we would appreciate any references that could enhance our related work section.
>
> We hope this clarifies the purpose and originality of our work. We proceed to answer the specific questions of Reviewer M9yc:

---

> ### Author Response · Authors · 2024-11-15
> **Continuation to Previous Response To Reviewer M9yc**
>
> __Q1. SCOTT.__ As noted, incorporating CNN priors to improve ViT tokenization is no new in supervised learning, but it presents distinct challenges in SSL MIM tasks. Focusing on Spark, which pioneered the idea of introducing sparse convolutions in MIM, their work proposes a fully convolutional encoder-decoder architecture reminiscent of UNet and train in a generative BERT style framework, this is far from our work in both architectural design (Fully CNN vs. ViT) and learning framework perspective (Generative Architecture vs. JEPA).
>
> While introducing SCOTT might now seem obvious after reading the paper to improve efficiency of ViTs this should not be taken for granted, to the best of our knowledge there is no prior work to propose a CNN-like tokenizer for ViTs that is compatible with MIM tasks, nor to prove its effectiveness. This is indeed our contribution, we are the __first__ to propose to replace the patch and embed strategy of ViTs by a shallow convolutional tokenizer that is compatible with MIM tasks to enable efficient training. This compatibility with any MIM framework (e.g., iBOT, I-JEPA, DinoV2) demonstrates SCOTT novelty and potentiality to improve the data-efficiency of standard ViT models trained in them.
>
> __Q2. MIM-JEPA.__ We appreciate the request for clarification on MIM-JEPA’s design. Although MIM-JEPA and I-JEPA both instantiate a Joint-Embedding Predictive Architecture (JEPA), their specific implementation differs, the most two salient:
> - __Context processing:__ In I-JEPA, only the visible patches are processed by the context-encoder. MIM-JEPA, in contrast, processes both visible and masked patches within the transformer part of the context-encoder, resulting in more extensive computation for representation learning for the masked areas.
> - __Predictor:__ In I-JEPA, the predictor receives the “encoded visible patches” and conditioned on positional tokens, predicts the representations of a target block at a specific location. In contrast, in MIM-JEPA, the predictor receives all patches, both visible and masked, and predicts the representations of their corresponding target patches.
>
> While both methods are JEPAs, the specific implementation details mark a difference in making it effective for small datasets.
>
> __Q3. Evaluation concerns:__
>
> __Q3.1. Supervised Baselines:__ as stated in Appendix “E.2. Evaluation Details” of the original manuscript: “Supervised ViTs and SCOTT variants are trained for 300 epochs”. We appreciate the opportunity to clarify this aspect and confirm that supervised baselines were trained from scratch with the entire model optimized over the same number of epochs as the pre-trained models. We hope this response clarifies Reviewer M9yc’s  confusion about the fairness of our experiments and we would like to take this opportunity to highlight probing on frozen features produced by our pre-training method outperforms by far all this fairly trained supervised baselines (>26% top-1 accuracy in Flowers102 and >38% top-1 in Pets37, extracted from Table 1 of the original manuscript).

---

> ### Author Response · Authors · 2024-11-15
> **Continuation to Previous Response To Reviewer M9yc**
>
> __Q3.2. Unsupervised Pre-Training on Target Datasets:__ we respectfully disagree with the notion that pre-training on the full, unlabeled target dataset creates an unfair advantage, as no label-based learning signal involved in this process. Our approach aligns with standard practices in the field of representation learning, where our work is situated, and differs fundamentally from supervised learning, where we suspect the concern about data overlap (“this overlap can lead to an advantage, as the model learns features directly from the target dataset”) may originate.
>
> In representation learning, it is standard practice to use the full, unlabeled dataset for pre-training as the model learns structural patterns within the data independently of labels. This practice contributes to efficient learning in data-limited settings without requiring large external datasets, which is precisely the focus of our work: to demonstrate that one can train a ViT from scratch with less than 10,000 images and achieve results comparable to those obtained by methods pretrained on massive datasets, such as the 142 million images in LVD-142M used to train DinoV2. To reinforce this point, we refer to Table 15 in DinoV2 original manuscript (“Composition of our LVD-142 Dataset”), which shows that Flowers102, Pets37, and ImageNet are all included in the process of achieving the LVD-142 Dataset.
>
> __Cross-dataset generalization evaluation__: while cross-dataset generalization is an interesting research direction, it is not the intended goal of our work and is unfair to judge it from that perspective. As Reviewer M9yc suggests “For instance, if Flowers-102 is used for evaluation, then Pets-37 or ImageNet-100 could serve as a pre-training dataset” is a relevant question for “foundational models” trained with extensive resources, but we do not claim such generalization capabilities from our method, which is focused on effective performance given limited data.
>
> Additionally, given the small sample sizes and fine-grained nature of the datasets used in our study, to the best of our knowledge, no existing methods have demonstrated strong cross-dataset generalization under these conditions. In particular, to the best of our knowledge it is still an unresolved problem to pre-train on a small, unrelated, fine-grained dataset and then perform well in an evaluation setting where the features differ significantly across domains (e.g., flowers vs. pets) -especially when probing, where simple classifiers are trained on top of frozen features.
>
> __Q4. Accuracy gap between method and SOTA.__ We hope that the extensive clarification on the fairness of our experiments will solve your concerns regarding the “promising” potential of SCOTT+MIM-JEPA, given our primary objective: to enable effective model training on small-scale, fine-grained datasets -a direction that “is critical for advancing self-supervised learning in data-limited settings” citing Reviewer  M9yc’s words in the strengths section.
>
> While pretraining on large-scale datasets or fine-tuning from open-source, pre-trained models may be the optimal approach in the natural image domain, our work addresses the need for effective SSL methods on small datasets, which is especially relevant for a broad range of applications (e.g., medical imaging, robotics) where large-scale data is unavailable or costly to obtain. We believe that training competitive models with constrained resources offers great value to a "long tail" of computer vision tasks that may not benefit from traditional large-scale pretraining.
>
> Regarding the noted “accuracy gap,” it is important to emphasize that our results were achieved using probing, where a simple classifier is trained on top of frozen features. Full fine-tuning would likely yield higher accuracy but is outside the scope of our current study which is not to achieve state-of-the-art and is left for future work.

---

> ### Author Response · Authors · 2024-11-15
> **Continuation to Previous Response To Reviewer M9yc**
>
> __Minor points:__
>
> __M1. Contrastive objectives:__ Our method, similar to other ViT approaches, can easily incorporate a CLS token to support contrastive objectives if desired. While contrastive learning methods, such as Dino or BYOL, have achieved impressive results, they generally rely on view-invariant transformations to optimize different crops of the same image to converge on a single “class” representation. This assumption is effective for datasets like ImageNet, where the main object is typically isolated and centered in the image.
>
> However, generalizing this assumption to more diverse, "in-the-wild" datasets is challenging, as these datasets often contain complex backgrounds or multiple objects without a single dominant subject. For further context, we refer to the insights on these limitations in Balestriero R. et al.  “A Cookbook of Self-Supervised Learning”. Therefore, while adding a CLS token for contrastive objectives is technically straightforward, our focus remains on methods better suited to the diversity of real-world data and less structured datasets.
>
> __M2 and M3.__ Improving reading experience. We will kindly hear suggestions on how we could present the information to enhance a better understanding of our unique contributions. As per the Reviewer M9yc’s minor point consideration of introducing I-JEPA in the background, we will do our best to add it while it is a challenge given the manuscript page limit policy.
>
> We hope this extensive clarification will help Reviewer M9yc provide an updated evaluation and informed judgement of our work, with a focus on our key contributions to escape the big data paradigm in computer vision and making it more accessible to a wider range of applications. We are looking forward to the reviewer’s response and appreciate their time and consideration.

---

### Meta-Review · Area_Chair_yW8V · 2024-12-20

**Metareview:**

This paper introduces SCOTT and MIM-JEPA for representation learning in the absence of large-scale datasets, where the former injects convolutional inductive biases to ViTs and the latter combines masked image modeling with JEPA for better semantic understanding. The resulting framework shows promising results in three small fine-grained tasks.

However, the novelty of the work is somewhat incremental as similar approaches have been proposed in the previous literature. The evaluations of the proposed method are limited to only a few small sets. Therefore, I would recommend rejecting the paper.

**Additional Comments On Reviewer Discussion:**

Reviewers M9yc, BkxW, and 6LsZ all point out the limited novelty and similarity to previous work. In particular, reviewers M9yc and 6LsZ mentioned that the results are not very comparable. Reviewer M9yc mentioned that the final result is still lower than the current best model, putting a question mark on the motivation. The authors replied to these justification questions by restating that the focus of the work is representation learning on small data. As the AC, I think the reviewers raised a valid point and this is the main reason for rejection.

In addition, reviewer 7fBC asked for more relevant work and experiments. The authors explained the setup and relevant works in detail, but the added experiments are within limited setups.

---

### Decision · Program_Chairs · 2025-01-22

Reject